# Bioactive triterpenoids from *Solanum torvum* fruits with antifungal, resistance modulatory and anti-biofilm formation activities against fluconazole-resistant *candida albicans* strains

Benjamin Kingsley Harley[ID][1], David Neglo[2], Philip Tawiah[ID][1], Mercy Adansi Pipim[1], Nana Ama Mireku-Gyimah[3], Clement Okraku Tettey[4], Cedric Dzidzor Amengor[5], Theophilus Christian Fleischer[1], Sayanika Devi Waikhom[ID][4]*

1 Department of Pharmacognosy and Herbal Medicine, School of Pharmacy, University of Health and Allied Sciences, Ho, Ghana, 2 Department of Basic Science, School of Basic and Biomedical Sciences, University of Health and Allied Sciences, Ho, Ghana, 3 Department of Pharmacognosy and Herbal Medicine, School of Pharmacy, University of Ghana, Accra, Ghana, 4 Department of Biomedical Science, School of Basic and Biomedical Sciences, University of Health and Allied Sciences, Ho, Ghana, 5 Department of Pharmaceutical Chemistry, School of Pharmacy, University of Health and Allied Sciences, Ho, Ghana

* swaikhom@uhas.edu.gh, sayanikawaikhom@gmail.com

## Abstract

Vulvovaginal candidiasis (VVC) is the second most common vaginal infection that affects women of reproductive age. Its increased occurrence and associated treatment cost coupled to the rise in resistance of the causative pathogen to current antifungal therapies has necessitated the need for the discovery and development of novel effective antifungal agents for the treatment of the disease. We report in this study the anti-*Candida albicans* activity of *Solanum torvum* 70% ethanol fruit extract (STF), fractions and some isolated compounds against four (4) fluconazole-resistant strains of *C. albicans*. We further report on the effect of the isolated compounds on the antifungal activity of fluconazole and voriconazole in the resistant isolates as well as their inhibitory effect on *C. albicans* biofilm formation. STF was fractionated using n-hexane, chloroform (CHCl₃) and ethyl acetate (EtOAc) to obtain four respective major fractions, which were then evaluated for anti-*C. albicans* activity using the microbroth dilution method. The whole extract and fractions recorded MICs that ranged from 0.25 to 16.00 mg/mL. From the most active fraction, STF-CHCl₃ (MIC = 0.25–1.00 mg/mL), four (4) known compounds were isolated as Betulinic acid, 3-oxo-friedelan-20α-oic acid, Sitosterol-3-β-D-glucopyranoside and Oleanolic acid. The compounds demonstrated considerably higher antifungal activity (0.016 to 0.512 mg/mL) than the extract and fractions and caused a concentration-dependent anti-biofilm formation activity. They also increased the sensitivity of the *C. albicans* isolates to fluconazole. This is the first report of 3-oxo-friedelan-20α-oic acid in the plant as well as the first report of betulinic acid, sitosterol-3-β-D-glucopyranoside and oleanolic acid from the fruits of *S. torvum*. The present study has demonstrated the anti-*C. albicans* activity of the constituents of *S. torvum* ethanol fruit extract and also shown that the constituents possess anti-biofilm

**Data Availability Statement:** All relevant data are within the paper and its Supporting information files.

**Funding:** The author(s) received no specific funding for this work.

**Competing interests:** The authors have declared that no competing interests exist.

formation and resistance modulatory activities against fluconazole-resistant clinical *C. albicans* isolates.

## 1 Introduction

Vulvovaginal candidiasis (VVC), the second most common vaginal infection affects about 70–75% reproductive-aged women worldwide [1]. This infection occurs due to the abnormal growth of *Candida* species in the mucous membrane of the female genital area with *Candida albicans* accounting for 85–90% of the fungal infection [2]. The condition can be recurrent and is associated with severe itching, dyspareunia and dysuria, vaginal discharge and vulvar erythema and oedema [3]. Although VVC is not life-threatening, it can lead to disruption of routine and daily social and occupational activities, neurological problems such as anxiety and depression, and even sexual complications [4]. The treatment options for vulvovaginal candidiasis are standard antifungal agents such as azoles, echinocandins and polyenes which are available in oral and/or vaginal formulations [5]. The azoles, particularly fluconazole and clotrimazole have emerged as the drugs commonly used in treating VVC due to their ease of administration and favourable pharmacokinetic profile. However, like other antifungal drugs, their extensive use has resulted in the emergence of resistant *C. albicans* rendering them ineffective in the treatment of VVC and leading to recurrent yeast infections in some cases [6, 7]. Additionally, vaginal clotrimazole has been shown to cause hypersensitivity reactions like burning, itching, erythema and some drug interactions [8]. The increased occurrence and associated treatment cost of VVC, in addition to the rise in the rate of resistance to current antifungal therapies has necessitated the need for the discovery and development of novel effective antifungal agents for the treatment of the disease.

*Solanum torvum* Swartz is a small Solanaceous tree with woody taproots that grows up to 5 m high. It is widely distributed in the tropical regions of the world and cultivated in Africa, Asia and the West Indies [9]. *S. torvum* has alternate leaves which are broadly ovate with entire or lobed margins [10]. The globular shaped edible fruits grow in clusters and have a bitter taste when cooked whereas the stems have scattered hooked prickles [11]. In Ghana, the leaves and fruits of *S. torvum* are boiled and used in the treatment of diabetes, hypertension, malaria and tuberculosis [12–14]. Leaf poultices are applied directly on skin lesions [15]. The fruit is also used a tonic and a haematinic for pregnant women and in the treatment of bacterial and fungal infections [16]. Phytochemical investigation of *S. torvum* has led to the isolation of several steroidal glycosides including steroidal alkaloids such as chlorogenone and neochlorogenone [17]. Torvosides A-M and sitosterol β-D-glucopyranoside were isolated from the leaves whereas astorvosides A-G were obtained from the roots [18].

The fruits of *S. torvum* have also yielded steroidal glycosides like 5α-pregn-16-en-3,20-dione-6α-ol-6-O-[α-L-rhamnopyranosyl-(1→3)-β-D-quinovopyranoside], 25(S)-26-O-β-D-glucopyranosyl-5α-furost-22(20)-en-3β,6α,26-triol-6-O-β-D-quinovopyranoside, 25(S)-26-O-β-D-glucopyranosyl-5α-furost-22(20)-en-3β,6α,26-triol-6-O-[α-L-rhamnopyranosyl-(1→3)-O-β-D-quinovopyranoside] and 5α-pregn-16-en-20-one-3β,6α-diol-6-O-[α-Lrhamnopyranosyl-(1→3)-β-D-quinovopyranoside] [19]; and phenolic compounds including methyl caffeate and isoflavonoid sulfate [20]. The fruits are reported to possess anti-ulcerogenic, anti-hypertensive, anti-inflammatory [10], antidiabetic [21], anticancer [22] and antibacterial activities [23].

Studies have also shown that the fruits of *S. torvum* possess antifungal activities against fluconazole-susceptible strains of *C. albicans* [24–26]. However, in those reports, the constituents responsible for the antifungal activity were neither isolated nor tested. The present study therefore sought to isolate and investigate the antifungal activity of the constituents of the fruits of *S. torvum* against fluconazole resistant clinical isolates of *C. albicans*.

## 2 Materials and methods

### 2.1 Drugs and chemicals

Fluconazole, voriconazole, chloramphenicol, Mueller-Hinton (MH) agar and Sabouraud Dextrose Agar (SDA) were obtained from Thermo Fisher (Oxoid Limited, Hampshire, UK). All the organic solvents were purchased from BDH laboratory supplies (Merck Ltd, Lutterworth, UK).

### 2.2 Plant material collection and processing

The fruits of *S. torvum* were bought from the local markets in Ho, Volta Region in August, 2020 and authenticated by Mr. Alfred Ofori at the Institute of Traditional and Alternative Medicine (ITAM), University of Health and Allied Sciences (UHAS) where voucher specimen has been deposited (Voucher specimen number: UHAS/ITAM/2021/FRCO2). The fruits were washed under running water, chopped into pieces, air-dried for a week and ground into coarse powder.

### 2.3 General procedures

Column Chromatography (CC) was performed using silica gel 60 (70–230 mesh; AppliChem, GmbH, Darmstadt, Germany) or sephadex LH-20 (25–100 μm; Amersham Biosciences) as stationary phases. Thin Layer Chromatography (TLC) was carried out using pre-coated silica gel 60 plates (0.25 mm thickness) incorporated with fluorescent indicator $GF_{254}$. 1D and 2D NMR spectra were recorded at 25°C on a Bruker Avance-500 (500 MHz). Chemical shifts (δ) were expressed in parts per million (ppm) using tetramethylsilane (TMS) as internal standard and coupling constants (*J*) were measured in Hertz (Hz).

### 2.4 Extraction, fractionation and isolation of compounds

The air-dried powdered fruits of *S. torvum* (2.7 kg) were extracted by cold maceration with 70% ethanol (3 x 3 days) at room temperature with occasional agitation. The combined extracts were then concentrated under reduced pressure using the rotary evaporator to obtain a solid extract (STF, 174 g). Thereafter, 150 g of STF was suspended in distilled water and partitioned successively with hexane, chloroform ($CHCl_3$) and ethyl acetate (EtOAc) and concentrated to obtain four respective major fractions: STF-hexane (8.1 g), STF-$CHCl_3$ (45.7 g), STF-EtOAc (11.4 g) and STF-AQ (68.9 g) fractions [27].

STF-$CHCl_3$ (32 g) was subjected to column chromatography on silica gel eluting with gradients of $CHCl_3$: MeOH (v/v, from 90:1 to 1:1) to obtain 5 fractions: F1 (0.8 g), F2 (7.0 g), F3 (0.6 g), F4 (2.8 g), (3.5 g) and F5 (6.6 g) based on the TLC profiling [28]. Purification of fraction F5 (6.6 g) [Pet ether-EtOAc—6:4 to 0:1] using silica gel column chromatography afforded 110 fractions (20 mL each) bulked into 3 sub-fractions, F5A, F5B and F5C according to their TLC profiles. Sub-fraction F5A was further column chromatographed on sephadex LH-20 with $CHCl_3$: MeOH (1:1) to yield compounds **1** (1.7 g, 0.063%) and **2** (1.3 g, 0.048%). Sub-fraction F5B was also column chromatographed on sephadex LH-20 using $CHCl_3$: MeOH (1:1) as the eluent to give compound **3** (0.94 g, 0.035%). Compound **4** (0.75 g, 0.028%) was obtained

after subjecting sub-fraction F5C to column chromatography on sephadex LH-20 using CHCl$_3$: MeOH (1:1) as the eluent [29].

### 2.5 Antifungal testing

**2.5.1 Fungal strains isolation and growth conditions.** Clinical isolates of *C. albicans* were obtained from the Microbiology laboratory of the Ho Teaching Hospital, Ho, Ghana. They were primarily isolated from pregnant women who reported with VVC and were resistant to antifungal agents [30]. To ensure the purity of the *Candida* isolates, separate yeast colonies were sub-cultured on Sabouraud Dextrose Agar (SDA) with chloramphenicol before incubation at 37˚C for 48 h. Identification of *C. albicans* was carried out by by culturing the isolates on HiCrome Candida Differential Agar (HiMedia Laboratories, India) at 35˚C for 48 h for production of species-specific colours. *C. albicans* isolates appeared as light green coloured smooth colonies and were selected for the study. Confirmation using API ID 32C strips (Biomerieux, France) were carried out according to standard microbiological methods [30].

**2.5.2 Fluconazole susceptibility test.** Antifungal susceptibility testing of fluconazole (25 μg) was carried out by disc diffusion on Mueller-Hinton (MH) agar method as described by [31] with slight modifications. Briefly, with a sterile inoculating loop, an inoculum was prepared using distinct colonies of *C. albicans* isolates from the SDA plates which were transferred into a 5 mL test tube containing 0.85% sterile saline solution, and emulsified to form a suspension of turbidity equivalent to 0.5 McFarland standard as compared to a 0.5 McF PhoenixSpec Calibrator (Becton, Dickinson and Company, USA). Thereafter, the media lawn was seeded in three dimensions using sterile swabs dipped in prepared inoculum. Then fluconazole-loaded disks were aseptically placed on the lawn and incubated at 37˚C for 24–48 h. The zone diameters of antifungal disks were determined using a measuring ruler. Zone diameter of $\geq$ 19 mm was considered sensitive, 15 to 18 mm dose-dependently susceptible, and $\leq$ 14 mm considered resistant. Subsequently, four strains of fluconazole resistant *C. albicans* isolates were used for the study and were designated as CA-1, CA-2, CA-3 and CA-4 respectively.

**2.5.3 Evaluation of antifungal activity.** The antifungal activity of STF and its major fractions; column fractions of the active STF-CHCl$_3$ and isolated compounds was tested using the microbroth dilution method based on the CLSI documents M27—A3 as previously described [32]. Fluconazole and voriconazole were included as positive controls and 0.1% DMSO employed as a negative control. The experiment was carried out in triplicate.

The plant extract, partitioned and column fractions were considered active when MIC was < 0.1 mg/mL, moderately active when MIC ranged from 0.1 to 0.5 mg/mL, and weakly active when MIC was from 0.5 to 1 mg/mL. Above 1 mg/mL they were considered inactive [33].

The antifungal activity of the isolated compounds was interpreted as follows: very strong bioactivity <3.515 μg/mL; strong bioactivity 0.003515–0.025 mg/mL; moderate bioactivity 0.026–0.10 mg/mL; weak bioactivity 0.101–0.5 mg/mL; very weak bioactivity 0.5–2 mg/mL; and no activity above 2 mg/mL [34].

The modulatory effect of the isolated compounds on either fluconazole or voriconazole was also determined. The Minimum Inhibitory Concentration (MIC) of fluconazole or voriconazole were determined in the absence or presence of the compounds (¼ MIC) respectively against the *C. albicans* isolates using the broth dilution procedure as earlier described. The modulatory factor (MF) which is a measure of the modulation effect of the compounds on the MIC of the antifungals was calculated as the ratio of the MIC of the antifungals alone to the MIC of the antifungals in the presence of the compounds.

**2.5.4 Checkerboard assay.** Interactions of the isolated compounds with either fluconazole or voriconazole were investigated using the broth microdilution checkerboard procedure modified from the EUCAST-AFST guidelines reference technique as previously described [35]. Final concentrations for all test samples ranged from 0.063 to 64 μg/mL. The mode of the interactions was measured by calculating the Fraction Inhibitory Concentration Index (FICI).

For calculation of the FICI, the FIC of each drug was first determined as follows: $FIC_A$ was obtained by dividing the MIC of drug A when used in combination ($MIC_{CA}$) by the MIC of the drug when used alone ($MIC_A$). Similarly, FIC of drug B was obtained by dividing the MIC of drug B when used in combination ($MIC_{CB}$) by the MIC of the drug when used alone ($MIC_B$).

FICI was then calculated as follows:

$$FICI = FIC\ A + FIC\ B = (MIC_{CA}/MIC_A) + (MIC_{CB}/MIC_B)$$

Interactions were interpreted as follows: Synergism for FICI $\leq 0.5$, Indifference FICI was $> 0.5$ to $\leq 4.0$, and Antagonism FICI $> 4.0$ [36].

**2.5.5 Biofilm inhibition assay.** The tendency of the isolated compounds to inhibit biofilm formation by the strains of fluconazole resistant *C. albicans* isolates was investigated using the microplate crystal violet stain retention assay according to a previously described method [6] with slight modifications. Briefly, each of the compounds was dissolved in 50 μL of the Mueller Hinton broth in 96-well plates to obtain concentrations of 100, 30, 10 and 3 μg/mL. Thereafter 10 μL each of the *C. albicans* isolates containing $10^5$ per mL microorganisms were added. As control, wells without compounds were included. The plates were then incubated at 37°C for 48 h after which the planktonic cells were aspirated and wells dried at 25°C. The attached cells were then stained with 0.1% crystal violet and incubated for 15 mins. The adherent microbial biofilm on the walls of each well was then reconstituted with 150 μL ethanol and the absorbance determined at 595 nm.

Percentage biofilm inhibition was calculated as follows:

$$\% \text{ Biofilm inhibition} = \frac{\text{Absorbance of control} - \text{Absorbance of treatment}}{\text{Absorbance of Control}} \times 100$$

Each strain and concentration were assayed in three wells on each plate. The experiment was also replicated thrice.

# 3. Results

## 3.1 Isolation and identification of compounds from the fruits of *S. torvum*

Phytochemical investigation of the partitioned $CHCl_3$ fraction of STF led to the isolation of four (4) known compounds. They were identified on the basis of their $^1H$ and $^{13}C$ NMR spectra and in comparison, to reported literature as Betulinic acid (**1**), 3-oxo-friedelan-20α-oic acid (**2**), Sitosterol-3-β-D-glucopyranoside (**3**) and Oleanolic acid (**4**) (Fig 1). This is the first report of 3-oxo-friedelan-20α-oic acid (**2**) in *S. torvum*. The spectral data of the compounds are provided in the supporting information.

## 3.2 Antifungal activity of *S. torvum* extract and partitioned fractions

STF and partitioned fractions were active against the *C. albicans* isolates to varying extent in the microbroth dilution assay with MIC ranging from 0.25 to 16 mg/mL (Table 1). STF demonstrated moderate to little activity against the fungal growth with MIC between 0.25 and 2 mg/mL. The most active fraction was STF-$CHCl_3$ with an MIC of 0.25–1.00 mg/mL. The least active fractions were STF-hexane and STF-AQ. All the isolates showed resistance to

**Fig 1. Structures of compounds isolated from the fruits of *S. torvum*.**

fluconazole with MIC >0.064 mg/mL. Voriconazole had variable inhibitory activities on the isolates (MIC = 0.004–0.016 mg/mL).

### 3.3 Antifungal activity of the column fractions of STF-CHCl₃

STF-CHCl₃ exhibited the highest antifungal activity among the fractions evaluated and was subsequently fractionated using column chromatography over silica gel to obtain five fractions

**Table 1. Antifungal activity of *S. torvum* ethanol fruit extract and fractions against clinical isolates of *C. albicans*.**

| Strain | Minimum Inhibitory Concentration (mg/mL) | | | | | | |
|---|---|---|---|---|---|---|---|
| | STF | STF-hexane | STF-CHCl₃ | STF-EtOAc | STF-AQ | FLC | VRC |
| CA-1 | 1.00 | 16.00 | 0.50 | 0.50 | 16.00 | >0.064 | 0.008 |
| CA-2 | 0.50 | 16.00 | 0.50 | 2.00 | 16.00 | >0.064 | 0.016 |
| CA-3 | 0.25 | 16.00 | 0.25 | 8.00 | 16.00 | >0.064 | 0.008 |
| CA-4 | 2.00 | 16.00 | 1.00 | 4.00 | 4.00 | >0.064 | 0.004 |

STF: Ethanol extract of *Solanum torvum* fruits; STF-hexane, STF-CHCl₃, STF-EtOAc and STF-AQ: hexane, chloroform, ethyl acetate and aqueous fractions respectively of ethanol extract of *S. torvum* fruits. FLC: Fluconazole; VRC: Voriconazole. Experiment was carried out in triplicate.

**Table 2. Antifungal activity of the column fractions of STF-CHCl₃ against clinical isolates of *C. albicans*.**

| Strain | Minimum Inhibitory Concentration (mg/mL) | | | | | | |
|--------|------|------|------|------|-------|--------|-------|
| | F1 | F2 | F3 | F4 | F5 | FLC | VRC |
| CA-1 | 1.00 | 2.00 | 0.50 | 4.00 | 0.125 | >0.064 | 0.008 |
| CA-2 | 16.00 | 16.00 | 16.00 | 16.00 | 0.125 | >0.064 | 0.016 |
| CA-3 | 16.00 | 4.00 | 8.00 | 16.00 | 0.125 | >0.064 | 0.008 |
| CA-4 | 8.00 | 8.00 | 4.00 | 8.00 | 0.250 | >0.064 | 0.004 |

F1, F2, F3, F4 and F5: Column fractions of the chloroform-partitioned fraction of *S. torvum* ethanol fruit extract (STF-CHCl₃). FLC: Fluconazole; VRC: Voriconazole. Experiment was carried out in triplicate.

(F1 –F5). The antifungal activities of the column fractions are presented in Table 2. They demonstrated varying effects on the *C. albicans* isolates with column fraction 5 (F5) exhibiting the greatest activity (MIC = 0.125–0.250 mg/mL) across the panel of isolates. The activity of voriconazole and fluconazole were as earlier indicated.

## 3.4 Antifungal activity of isolated compounds

The antifungal activity of the compounds against the growth of the *C. albicans* isolates are displayed in Table 3. The compounds demonstrated antifungal activity at MIC from 0.016 to 0.512 mg/mL. 3-oxo-friedelan-20α-oic acid (**2**) recorded the highest inhibitory effects with MIC of 0.016 and 0.032 mg/mL against CA-1 and CA-2; and CA-3 and CA-4 respectively, followed by betulinic acid (**1**) [MIC = (0.032 mg/mL against CA-1 and CA-4) and (0.064 mg/mL against CA-2 and CA-3)] and oleanolic acid (**4**) [MIC = (0.032 mg/mL against CA-1) and (0.064 mg/mL against CA-2, CA-3 and CA-4)]. The least potent compound was sitosterol-3-β-D-glucopyranoside (**3**) with MIC of 0.512 mg/mL against the isolates (Table 3).

## 3.5 Effect of the isolated compounds on the antifungal activity of fluconazole and voriconazole

**3.5.1 Modulation effect.** The modulatory effect of the isolated compounds at sub-inhibitory concentrations (¼ MIC) on the antifungal activity of fluconazole or voriconazole were investigated against the clinical isolates of *C. albicans*. The compounds affected the susceptibility of the *C. albicans* isolates towards fluconazole, albeit to varying extents. 3-oxo-friedelan-20α-oic acid (**2**) was the most effective modulator causing a considerable increase in the susceptibility of the *C. albicans* isolates to fluconazole (MF = > 16). The isolated compounds, however, reduced drastically the activity of voriconazole (Table 4).

**Table 3. Antifungal activity of isolated compounds against clinical isolates of *C. albicans*.**

| Strain | Minimum Inhibitory Concentration (mg/mL) | | | | | |
|--------|-------|-------|-------|-------|---------|-------|
| | 1 | 2 | 3 | 4 | FLC | VRC |
| CA-1 | 0.032 | 0.016 | 0.512 | 0.032 | >0.0 64 | 0.008 |
| CA-2 | 0.064 | 0.016 | 0.512 | 0.064 | > 0.064 | 0.016 |
| CA-3 | 0.064 | 0.032 | 0.512 | 0.064 | > 0.064 | 0.008 |
| CA-4 | 0.032 | 0.032 | 0.512 | 0.064 | > 0.064 | 0.004 |

FLC: Fluconazole; VRC: Voriconazole. Experiment was carried out in triplicate.

**Table 4. Minimum Inhibitory Concentration (MIC) of fluconazole and voriconazole in the absence or presence of the isolated compounds at ¼ MIC concentration against clinical isolates of *C. albicans*.**

| Test Sample | CA-1 | | CA-2 | | CA-3 | | CA-4 | |
|---|---|---|---|---|---|---|---|---|
| | MIC (mg/mL) | MF | MIC (mg/mL) | MF | MIC (mg/mL) | MF | MIC (mg/mL) | MF |
| **FLC** | >0.064 | | >0.064 | | >0.064 | | >0.064 | |
| **FLC + 1** | 0.008 | >8 | 0.016 | >4 | 0.008 | >8 | 0.008 | >8 |
| **FLC + 2** | 0.004 | >16 | 0.004 | >16 | 0.004 | >16 | 0.004 | >16 |
| **FLC + 3** | 0.008 | >8 | 0.016 | >4 | 0.016 | >4 | 0.032 | >2 |
| **FLC + 4** | 0.008 | >8 | 0.004 | >16 | 0.008 | >8 | 0.008 | >8 |
| **VRC** | 0.008 | | 0.016 | | 0.008 | | 0.004 | |
| **VRC + 1** | 0.032 | 0.25 | 0.016 | 1.00 | 0.016 | 0.50 | 0.008 | 0.50 |
| **VRC + 2** | 0.032 | 0.25 | 0.064 | 0.25 | 0.016 | 0.5 | 0.032 | 0.13 |
| **VRC + 3** | 0.064 | 0.13 | 0.016 | 1.00 | 0.064 | 0.13 | 0.016 | 0.25 |
| **VRC + 4** | 0.032 | 0.25 | 0.016 | 1.00 | 0.064 | 0.13 | 0.064 | 0.06 |

FLC: Fluconazole; VRC: Voriconazole. Modulation factor (MF) = MIC (FLC or VRC)/MIC (FLC or VRC + modulator); n = 3.

**3.5.2 Checkerboard assay.** The interactions of the antifungal combinations of the isolated compounds with fluconazole or voriconazole assessed using the checkerboard assay are summarised in Table 5. The compounds did not induce any synergistic action with fluconazole in the clinical strains of *C. albicans* with the exception of compound **2** with an FIC Index of 0.38 against CA-1. Their combinations with voriconazole, however, resulted in mostly antagonistic action.

## 3.6 Inhibition of biofilm formation

The isolated compounds demonstrated a concentration-dependent inhibition on biofilm formation in the fluconazole resistant clinical isolates of *C. albicans*(Fig 2). The percentage biofilm inhibition of the compounds ranged from 21 to 79%. Compound **2** gave the highest biofilm inhibitory effect against the isolates followed by compounds **4** and **1** respectively. CA-3 was the least susceptible to the test compounds.

**Table 5. Effect of the combined antifungal activity of isolated compounds and fluconazole or voriconazole by the checkerboard microbroth dilution assay.**

| Combinations | CA-1 | | CA-2 | | CA-3 | | CA-4 | |
|---|---|---|---|---|---|---|---|---|
| | FICI | INT. | FICI | INT. | FICI | INT. | FICI | INT. |
| **With FLC** | | | | | | | | |
| 1 | 0.63 | I | 0.63 | I | 1.13 | I | 4.06 | A |
| 2 | 0.38 | S | 1.13 | I | 0.63 | I | 2.13 | I |
| 3 | 1.13 | I | 1.13 | I | 2.06 | I | 4.06 | A |
| 4 | 0.63 | I | 1.13 | I | 1.13 | I | 2.13 | I |
| **With VRC** | | | | | | | | |
| 1 | 6.00 | A | 4.25 | A | 4.25 | A | 5.00 | A |
| 2 | 6.00 | A | 2.00 | I | 9.00 | A | 9.00 | A |
| 3 | 8.00 | A | 3.00 | I | 4.25 | A | 4.00 | I |
| 4 | 5.00 | A | 1.50 | I | 2.50 | I | 4.00 | I |

FLC: Fluconazole; VRC: Voriconazole; INT: Interpretation, FICI: Fraction Inhibitory Concentration Index. S: Synergism for FICI ≤0.5, I: Indifference FICI was >0.5 to ≤4.0, and A: Antagonism FICI >4.0.

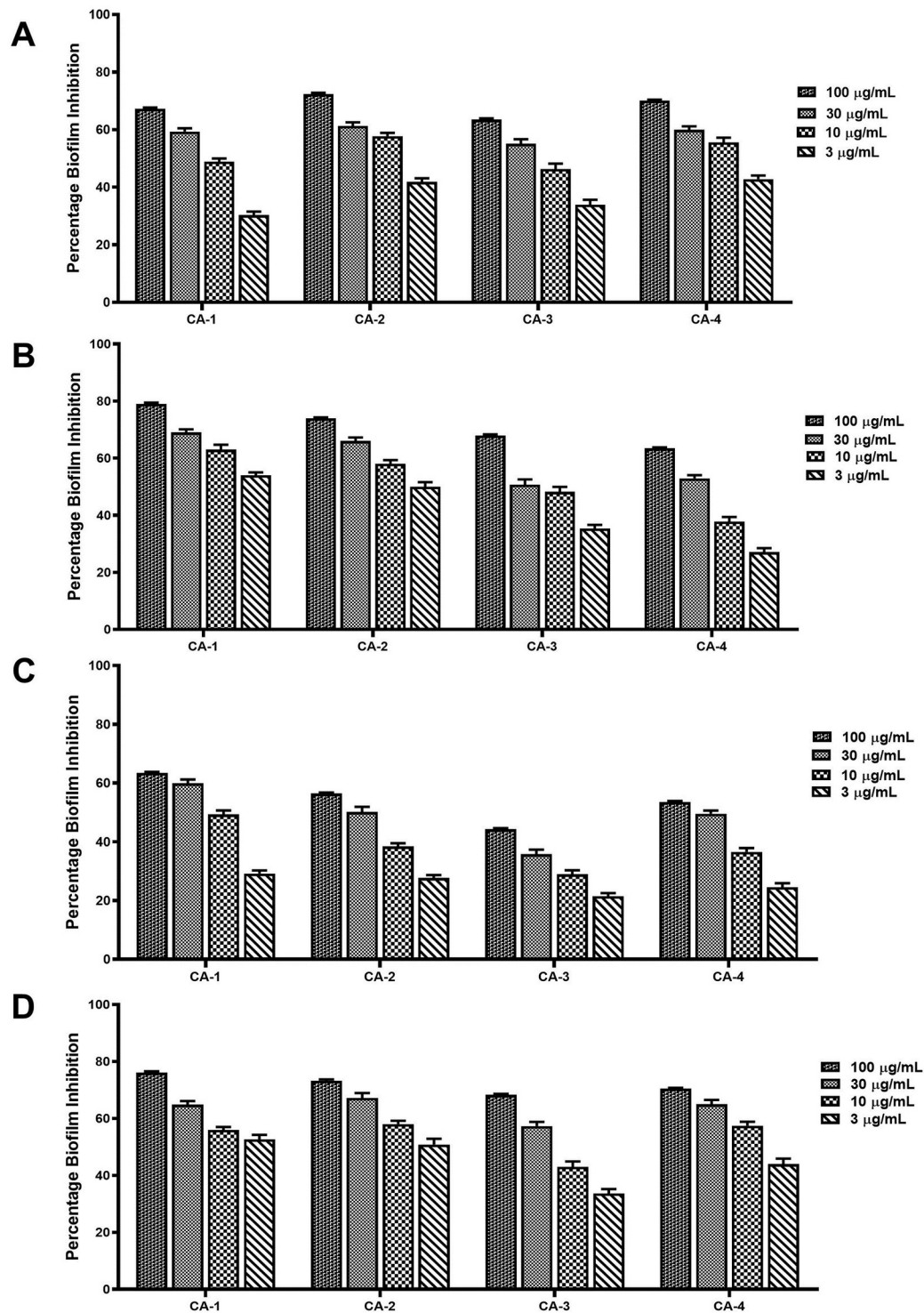

**Fig 2. Biofilm formation inhibitory effect of compounds 1–4 (A-D) in the Clinical isolates of *C. albicans* (CA-1, CA-2, CA-3 and CA-4).**

## 4. Discussion

STF demonstrated moderate to weak antifungal activity against the fluconazole-resistant clinical isolates with MIC ranging from 0.25 to 2.00 mg/mL [33]. The antifungal activity of STF in this study corroborates the findings of Obiang *et al* [25] who showed that the ethanol fruit extract of *S. torvum* possess antifungal activity against fluconazole-sensitive reference strains (ATCC 10231 and ATCC 90028) and clinical isolates of *C. albicans*. Evaluation of the solvent-partitioned fractions of STF revealed that its antifungal activity resided in the chloroform fraction (STF-CHCl₃). Whereas the other fractions gave very high MICs (2.00–16.00 mg/mL) indicating no antifungal activity, STF-CHCl₃ recorded low MIC values of 0.25 mg/mL against CA-3, 0.50 mg/mL against CA-1 and CA-2; and 1.00 mg/mL against CA-4 respectively indicating moderate activity which also compared well with the whole fruit extract (Table 1).

Following a bioassay-guided fractionation of STF-CHCl₃, four (4) known compounds were isolated from the fruits of *S. torvum*. They were obtained from the most active column fraction, F5 (MIC = 0.125–0.250 mg/mL) and identified based on their NMR analyses and by comparing their spectral data to reported literature as betulinic acid (**1**) [37], 3-oxo-friedelan-20α-oic acid (**2**) [38], sitosterol-3-β-D-glucopyranoside (**3**) [39] and oleanolic acid (**4**) [40]. To the best of our knowledge, this is the first-time report of Betulinic acid (**1**), sitosterol-3-β-D-glucopyranoside (**3**) and oleanolic acid (**4**) from the fruits of *S. torvum*, having been previously isolated from its aerial parts [41, 42]. Betulin was shown to be present in the fruits [43], however, its oxidative derivative betulinic acid was not detected. We are also reporting the presence of 3-oxo-friedelan-20α-oic acid (**2**) in the plant, *S. torvum* for the first time, although the compound has previously been reported in a number of plant species [44–46].

With the exception of sitosterol-3-β-D-glucopyranoside (**3**) (MIC of 512 μg/mL against all isolates), all the isolated compounds demonstrated greater antifungal activity compared to the fraction they were isolated from with MIC that ranged from 16 to 64 μg/mL. The strong to moderate antifungal activity of the compounds which were greater compared to their "parent" fraction suggests that the bioassay-guided fractionation and purification of the plant extract led to the isolation of some of the potent antifungal constituents of *S. torvum*.

Betulinic acid (**1**) has previously been reported to inhibit the growth of fluconazole-sensitive strains of *C. albicans* at MIC of 16 μg/mL [47]. Elsewhere, the compound effectively inhibited *C. albicans* secreted aspartic proteases (SAP), one of the most virulent factors in candida infection, at 6.5 μg/mL [48]. The low MIC of betulinic acid (**1**) against the fluconazole-resistant *C. albicans* isolates [(32 μg/mL against CA-1 and CA-4) and (64 μg/mL against CA-2 and CA-3)] reported in this study substantiates its potent antifungal activity. Favel *et al* [49] reported that triterpenoid saponins with oleanolic acid (**4**) as the aglycone demonstrated moderate antifungal activity whiles Eloff and co-workers [50] identified oleanolic acid (**4**) as the major antifungal constituent of *Melianthus comosus* against some plant pathogenic fungi through a bioactivity-guided isolation study. Oleanolic acid (**4**) and its oxime ester derivatives were also shown to inhibit *C. albicans* Glucosamine-6-phosphate synthase as a possible mechanism of antifungal activity [51]. The considerable antifungal activity of oleanolic acid (**4**) against the fluconazole-resistant *C. albicans* isolates [MIC = (32 μg/mL against CA-1) and (64 μg/mL against CA-2, CA-3 and CA-4)] may have also occurred through the same mechanism of action.

The antifungal activity of the five constituents including sitosterol-3-β-D-glucopyranoside (**3**), isolated from the stem of *Jatropha maheshwarii* were evaluated through the disc diffusion method. The result showed the constituents had weak antifungal activity in comparison to the whole extract and through a synergistic or additive effect, contributed to the activity of the *J. maheshwarii* stem [52]. Thus, the weak antifungal activity observed in the present study for sitosterol-3-β-D-glucopyranoside (**3**) lends support to that study.

This is the first report of the antifungal activity of 3-oxo-friedelan-20α-oic acid (**2**). It demonstrated the highest growth inhibitory effect [(16 μg/mL against CA-1 and CA-2) and (32 μg/mL against CA-3 and CA-4)] against the *C. albicans* isolates lending credence to the anti-*C. albicans* activity of friedelane-type triterpenoids [53].

Antifungal combination therapy (ACT) has been suggested as a promising strategy in overcoming resistance in recurrent candida infections. Again, the combination of antifungal agents with different mechanisms of action can increase efficacy and extend the spectrum of activity of the combined agents leading to synergism [54]. Furthermore, combination of different antifungal compounds can also reduce toxicity by reducing doses and improve pharmacokinetic profiles of one or both agents [3]. Hence, we investigated the antifungal combination activity of the isolated compounds with fluconazole or voriconazole by the checkerboard microbroth dilution assay. The combination of fluconazole with 3-oxo-friedelan-20α-oic acid (**2**) resulted in a synergistic action (FICI = 0.38) against CA-1, however, the predominant interaction observed in the *C. albicans* isolates for the combinations of the isolated compounds with the drug is indifferent. Even though the compounds failed to produce any synergistic effect against the clinical strains of *C. albicans* with fluconazole, at sub-inhibitory concentrations (¼ MIC), they significantly potentiated the antifungal effect of fluconazole (MF ranging from > 2 to > 16 folds) against them. The modulatory effect of natural products has been suggested to occur by their alteration of fungal cell wall to facilitate the access of antifungal agents to the fungal membrane or by inhibition of fungal efflux pumps [35]. The combinations of the isolated compounds with voriconazole mostly resulted in antagonism. This was further buttressed in the modulatory assay where MIC of voriconazole against the *C. albicans* isolates was increased by 2–16 folds indicating reduced antifungal activity.

*C. albicans* is a robust biofilm-forming organism which presents serious therapeutic complications [55]. Its ability to form biofilms contributes greatly to its resistance and recurrent infections [56]. With the limited therapeutic options for *Candida* biofilms available, there is a pressing need for the discovery and development of new anti-*C. albicans* biofilm agents. The isolated compounds demonstrated considerable concentration-dependent inhibition of biofilm formation in the *C. albicans* isolates (Fig 2). A number of plant constituents particularly phenolics and triterpenoids have been shown to demonstrate anti-candida biofilm activity by impairing adhesion of planktonic cells on abiotic surfaces, reducing biofilm metabolic activity or promoting biomass degradation of mature biofilms [57]. We report for the first time the anti-biofilm formation of the isolated compounds in *C. albicans*. Some of these compounds however, have been known to exert this effect in some bacteria. Betulinic acid (**1**) significantly decreased biofilm formation in *Pseudomonas aeruginosa* at sub-MIC levels and have also been shown to inhibit biofilm formation in *Cryptococcus neoformans* [58, 59] whiles oleanolic acid (**4**) has demonstrated anti-biofilm activity across a wide range of Gram-positive and Gram-negative bacteria [60, 61]. The anti-biofilm activity of the isolated compounds may contribute in part to the efficacy of *S. torvum* fruits in the treatment of recurrent vaginal infections caused by *C. albicans*.

The present study has demonstrated the anti-*C. albicans* activity of the constituents of *S. torvum* ethanol fruit extract and shown that the constituents possess anti-biofilm formation and resistance modulatory activities against fluconazole-resistant clinical *C. albicans* isolates.

## 5. Conclusion

In this study, we report the antifungal activity of *S. torvum* fruit extract, fractions and some constituents on the growth of fluconazole-resistant strains of *C. albicans*. The compounds demonstrated considerably higher antifungal activity than the extract and fractions and caused

a concentration-dependent anti-biofilm formation activity. In particular, 3-oxo-friedelan-20α-oic acid (**2**) reported in the plant for the first time significantly decreased the fluconazole-resistance levels of the *C. albicans* isolates.

## Supporting information

**S1 Fig. ¹H NMR spectrum of compound 1.**
(PDF)

**S2 Fig. DEPT Q NMR spectrum of compound 1.**
(PDF)

**S3 Fig. COSY spectrum of compound 1.**
(PDF)

**S4 Fig. HSQC spectrum of compound 1.**
(JPG)

**S5 Fig. HMBC spectrum of compound 1.**
(PDF)

**S6 Fig. ¹H NMR spectrum of compound 2.**
(PDF)

**S7 Fig. ¹³C NMR spectrum of compound 2.**
(PDF)

**S8 Fig. DEPT 135 spectrum of compound 2.**
(PDF)

**S9 Fig. HSQC spectrum of compound 2.**
(PDF)

**S10 Fig. HMBC spectrum of compound 2.**
(PDF)

**S11 Fig. ¹H NMR spectrum of compound 3.**
(PDF)

**S12 Fig. ¹³C NMR spectrum of compound 3.**
(PDF)

**S13 Fig. DEPT 135 spectrum of compound 3.**
(PDF)

**S14 Fig. HSQC spectrum of compound 3.**
(PDF)

**S15 Fig. HMBC spectrum of compound 3.**
(PDF)

**S16 Fig. ¹H NMR spectrum of compound 4.**
(PDF)

**S17 Fig. DEPTQ spectrum of compound 4.**
(PDF)

**S18 Fig. COSY spectrum of compound 4.**
(PDF)

**S19 Fig. HSQC spectrum of compound 4.**
(PDF)

**S20 Fig. HMBC spectrum of compound 4.**
(PDF)

**S1 File.**
(PDF)

## Acknowledgments

We thank Dr. Daniel A. Abaye for providing his laboratory for the phytochemical investigations and the Microbiology laboratory of the Ho teaching Hospital for providing the *C. albicans* clinical isolates.

## Author Contributions

**Conceptualization:** Benjamin Kingsley Harley, Sayanika Devi Waikhom.

**Data curation:** Benjamin Kingsley Harley, David Neglo, Sayanika Devi Waikhom.

**Formal analysis:** Benjamin Kingsley Harley, Nana Ama Mireku-Gyimah, Cedric Dzidzor Amengor, Sayanika Devi Waikhom.

**Investigation:** Benjamin Kingsley Harley, David Neglo, Philip Tawiah, Mercy Adansi Pipim.

**Methodology:** Sayanika Devi Waikhom.

**Project administration:** Benjamin Kingsley Harley.

**Supervision:** Benjamin Kingsley Harley, Clement Okraku Tettey, Theophilus Christian Fleischer, Sayanika Devi Waikhom.

**Validation:** Benjamin Kingsley Harley.

**Writing – original draft:** Benjamin Kingsley Harley, David Neglo, Nana Ama Mireku-Gyimah, Sayanika Devi Waikhom.

**Writing – review & editing:** Benjamin Kingsley Harley, Nana Ama Mireku-Gyimah, Clement Okraku Tettey, Cedric Dzidzor Amengor, Theophilus Christian Fleischer, Sayanika Devi Waikhom.

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
