## [Decision Letter · Decision Letter 0]

19 Aug 2021

PONE-D-21-19484

Bioactive triterpenoids from Solanum torvum fruits with Antifungal, Resistance Modulatory and Anti-Biofilm Formation Activities against Fluconazole-Resistant Candida albicans strains

PLOS ONE

Dear Dr. Waikhom,

Thank you for submitting your manuscript to PLOS ONE. After careful consideration, we feel that it has merit but does not fully meet PLOS ONE’s publication criteria as it currently stands. Therefore, we invite you to submit a revised version of the manuscript that addresses the points raised during the review process.

As mentioned by Reviewer 1, your study needs a significant revision. Please provide a point-by-point letter addressing the concerns of the reviewer.  In addition, please provide the following information:

1. The name of the botanist that identified the plant should be stated.

2. In Section 2.4, please add the percentage on the compounds following the weight.

3. Tables 1 and 2 expressed the results in mg/mL, whereas Table 3 in ug/mL. Please standarize the dimensions.

4. The word triplicate is not plural.

5. Please add a statistical analysis in Fig. 2.

We look forward to receiving your revised manuscript.

Kind regards,

Horacio Bach

Academic Editor

PLOS ONE

Journal Requirements:

Reviewers' comments:

Reviewer's Responses to Questions

**Comments to the Author**

1. Is the manuscript technically sound, and do the data support the conclusions?

Reviewer #1: Partly

Reviewer #2: Yes

2. Has the statistical analysis been performed appropriately and rigorously? 

Reviewer #1: N/A

Reviewer #2: N/A

3. Have the authors made all data underlying the findings in their manuscript fully available?

Reviewer #1: Yes

Reviewer #2: Yes

4. Is the manuscript presented in an intelligible fashion and written in standard English?

Reviewer #1: Yes

Reviewer #2: Yes

5. Review Comments to the Author

Reviewer #1: It is an interesting study but requires major changes before being accepted for publication

Vulnerable points:

The whole (crude) is not a promising antifungal since the MIC were light, so the entire manuscript needs to be revised.

Why has the study focused just on C. albicans isolates? It needs justification, as the non-Candida albicans species have increased in the last decades.

To draw relevant conclusions from just four C. albicans isolates, in my opinion, is very bold.

The English should be improved, maybe at the discretion of the editor.

Specific points

Abstract

It is missing results details regarding the antifungal activity,

What do the numbers in parentheses written after the compounds mean? Please delete this

The obtained results did not support the conclusion “justifying its use as an antifungal agent in the treatment of vulvovaginal candidiasis….” Lines 21-22

Introduction

Line 4: rephrase this sentence

Line 27: delete the word “ opportunistic”

Line 29: replace "The condition" by VVC

Line 35: the reference [5] is unsuitable for this statement, please replace it.

Line 62: Add the words of S. torvum after “The fruits”

Objectives: Rewrite it focusing on the purified compounds (isolation and their antifungal activity) it is the most relevant in the present study.

Methods

Lines 98-101: The authors should supply more details on the purification of Fraction F5, originating the compounds, also cite a reference, and so on

Lines 106-107: The growth in Sabouraud Dextrose Agar doesn't ensure the purity of the Candida isolates since this culture medium does not differentiate the colonial morphology between the species of this genus.

Line 116: it is written "ml", while elsewhere, as in line 142 it is "mL", standardize it in the entire manuscript, please

Line 126: the definition of the acronym STF .... ethanol extract (STF) is repeated other times, for instance, line 184 and others, please from the first definition (line 116) mention only STF

Line 155: This experiment is confusing, if the crystal violet test was performed in microplates, why were the compounds diluted in test tube sets? Next, line 159 again "The tubes were then incubated at …..", please rewrite this paragraph, clarifying the biofilm inhibition assay.

Results

Line 171: Is it missing some letters in the first word?

Fig 1: the caption is incomplete, it needs to be more informative, mentioning the compound names here instead of putting numbers every time compounds are cited in the text, as it makes reading confusing and tiring

Line 191: add the information “ethanol extract” into the title of table 1

Lines 185-187: The results are confusing, are the two sentences contradictory? Or is the second one just an interpretation? If yes, please move the sentence “STF was considerably active against the fungal growth with MIC between 0.25 and 2 mg/mL” to the methods session.

Line 229: Is the sentence “The isolated compounds, however, drastically reduced the activity of voriconazole” correct? I think it increased the activity of voriconazole Rephrase it, please

Discussion

The first paragraph should be deleted because it is confusing for instance (why was cited the reference 29?), and not relevant.

Line 271: What does “ considerable antifungal activity” mean? Please mention the reference used to define the breaking points in the interpretation of your results

Line 277: I disagree with the statement "low MIC values", as according to a recent systematic review by Alves et.al 2021 (https://doi.org/10.1155/2021/6653311), the values presented in Tables 1 and 2 (crude extract and fractions respectively) reveal only weak or no bioactivity. Please, clarify and discuss your data.

Line 342: This statement is true, but it needs a reference

Lines 360-361: The sentence “The reported activity justifies its use as an antifungal agent in the treatment of vulvovaginal candidiasis among the Ghanaian populace” must be deleted since it is not supported by data of the current study.

Conclusion

Lines 368-371: This subject is not a conclusion of the current study, maybe is possible to include it in the discussion session

Reviewer #2: Dear authors,

Solanum torvum is endemic in southern Mexico and South America, it is a widely studied plant. In their work they propose a possible therapeutic tool for antifungal treatment, making it a valuable proposal.

Best regards

6. PLOS authors have the option to publish the peer review history of their article (what does this mean?). If published, this will include your full peer review and any attached files.

Reviewer #1: **Yes: **Terezinha Inez Estivalet Svidzinski

Reviewer #2: No

---

## [Author Response · Author response to Decision Letter 0]

23 Oct 2021

The Editor-In-Chief

PLOS ONE

Cambridge, UK

Dear Madam,

RESPONSE TO REVIEWERS’ COMMENTS

We hereby in this letter provide responses to the queries raised by the academic editor and reviewers of our manuscript titled “Bioactive triterpenoids from Solanum torvum fruits with Antifungal, Resistance Modulatory and Anti-Biofilm Formation Activities against Fluconazole-Resistant Candida albicans strains” submitted to your revered journal PLOS One. All datasets generated for this study are included in the manuscript and/or the supplementary files.

Kindly find below our responses.

We hope it will receive a favourable consideration.

Thank you. 

Yours faithfully,

Sayanika Devi Waikhom (Ph.D)

(Corresponding Author)

Academic Editor 

1. The name of the botanist that identified the plant should be stated.

We have included the name of the Botanist who identified the plant.

Section 2.2, line 77

2. In Section 2.4, please add the percentage on the compounds following the weight.

The percentage weight of the compounds per weight of the dried powdered material have been included.

Section 2.4, lines 103-105

3. Tables 1 and 2 expressed the results in mg/mL, whereas Table 3 in ug/mL. Please standardize the dimensions.

The units of the all the tables have been standardized to mg/mL

Section 3.4 and Table 3.

4. The word triplicate is not plural.

The spelling of triplicate has been corrected.

Lines 136, 207, 220 and 232

5. Please add a statistical analysis in Fig. 2.

The analysis we conducted on the figure 2 was to determine the percentage biofilm inhibition. Since we are not comparing the inhibition to a control. we cannot carry out any further statistical analysis other than comparing the percentage inhibitions of the compounds to each other in terms of which one was higher and lower.

Reviewer 2

General comment

1. Why has the study focused just on C. albicans isolates? It needs justification, as the non-Candida albicans species have increased in the last decades.

We agree to the fact that non-albicans candida species have increased in the recent years, however we decided to focus on C. albicans in this study because despite the increase in the other species of candida, still Most Candida infections in

people are caused by Candida albicans [1]. The group however in other studies are investigating the effect of natural products on NAC such as C. glabrata and auris. We tried to justify the reason for our study on C. albicans in lines 28-29 and we provided an additional reference to support our justification.

2. To draw relevant conclusions from just four C. albicans isolates, in my opinion, is very bold.

We selected the four C. albicans strains as they covered a range of diameters considered as fluconazole-resistant which are ≤ 14 mm.

3. The English should be improved, maybe at the discretion of the editor.

We have improved the general English of the paper.

ABSTRACT

1. It is missing results details regarding the antifungal activity

We have included details regarding the antifungal activity in the abstract

Line 13-16

2. What do the numbers in parentheses written after the compounds mean? Please delete this

The number in parentheses represent the number assigned to the compounds in figure 1. This was done so readers could match the structure of the compound to the name. We have however deleted this as requested.

3. The obtained results did not support the conclusion “justifying its use as an antifungal agent in the treatment of vulvovaginal candidiasis….” Lines 21-22

The conclusion of the abstract has been re-written focusing on the isolated compounds. The statement “justifying its use as an antifungal agent in the treatment of vulvovaginal candidiasis….” has been deleted.

Line 20-23

INTRODUCTION

1. Line 4: rephrase this sentence

Unclear how the reviewer wants this sentenced rephrased.

2. Line 27: delete the word “ opportunistic”

“Opportunistic” has been deleted

Line 28

3. Line 35: the reference [5] is unsuitable for this statement, please replace it.

Appropriate reference has inserted

Reference [5]

4. Line 62: Add the words of S. torvum after “The fruits”

Line 62: Add the words of S. torvum after “The fruits”

Line 57

OBJECTIVES

Rewrite it focusing on the purified compounds (isolation and their antifungal activity) it is the most relevant in the present study.

The objectives have been re-written to focus on the isolation of the purified compounds and their subsequent testing.

Line 67-69

METHODS

1. Lines 98-101: The authors should supply more details on the purification of Fraction F5, originating the compounds, also cite a reference, and so on

Further details have been supplied on the purification of fraction F5.

Line 99-106

2. Lines 106-107: The growth in Sabouraud Dextrose Agar doesn't ensure the purity of the Candida isolates since this culture medium does not differentiate the colonial morphology between the species of this genus.

We have included how we identified and selected the C. albicans isolate

Line 113-116

3. Line 116: it is written "ml", while elsewhere, as in line 142 it is "mL", standardize it in the entire manuscript, please

The unit has been standardized and written as mL throughout the entire manuscript

4. Line 126: the definition of the acronym STF .... ethanol extract (STF) is repeated other times, for instance, line 184 and others, please from the first definition (line 116) mention only STF

The repeated definition of the acronym STF in the text has been deleted.

Line 132, 182, 195, 197, 277, 278, 281

5. Line 155: This experiment is confusing, if the crystal violet test was performed in microplates, why were the compounds diluted in test tube sets? Next, line 159 again "The tubes were then incubated at …..",please rewrite this paragraph, clarifying the biofilm inhibition assay.

The paragraph and the experiment have been re-written with clarity.

Section 2.5.5 lines 167-174

RESULTS 

1. Line 171: Is it missing some letters in the first word?

The missing letter has been included

Line 182

2. Fig 1: the caption is incomplete, it needs to be more informative, mentioning the compound names here instead of putting numbers every time compounds are cited in the text, as it makes reading confusing and tiring

As we stated earlier, the numbers before the names of the compounds are there to connect their structures to their names in fig 1. 

Using the numbers when the compounds are mentioned is a conventional practice of dealing with isolated compounds as seen in a number of journals including PLOS One. An example is this publication [2]. The caption of fig 1 has been modified.

Figure 1

3. Line 191: add the information “ethanol extract” into the title of table 1

We have added the ethanol to the information relating to the title of table 1

Line 202

4. Lines 185-187: The results are confusing, are the two sentences contradictory? Or is the second one just an interpretation? If yes, please move the sentence “STF was considerably active against the fungal growth with MIC between 0.25 and 2 mg/mL” to the methods session.

The first statement was a general summary on table 1 focusing on STF and its fractions. The second statement highlights the activity of STF alone before the 3rd statement finalizes on the most active fraction. Since it is a result, we do not think it will be appropriate to move it to the methods sections.

5. Line 229: Is the sentence “The isolated compounds, however, drastically reduced the activity of voriconazole” correct? I think it increased the activity of voriconazole Rephrase it, please

The statement is correct as MIC values correlates indirectly with antimicrobial activity. The lower the MIC value the higher the activity. From the results, the MIC of voriconazole was lower than when it was in combination with the isolated compounds therefore voriconazole alone had a higher activity than when it was combined with the compounds. It is for this reason we indicated that isolated compounds reduced the activity of the voriconazole

DISCUSSION

1. The first paragraph should be deleted because it is confusing for instance (why was cited the reference29?), and not relevant.

The first paragraph served as an introduction to the discussion. We have however deleted

2. Line 271: What does “considerable antifungal activity” mean? Please mention the reference used to define the breaking points in the interpretation of your results

We have re-written that aspect of the discussion and provided reference used to define the breaking points. We have also indicated that in the methods sections

Line 278-279

3. Line 277: I disagree with the statement "low MIC values", as according to a recent systematic review byAlves et.al 2021 (https://doi.org/10.1155/2021/6653311), the values presented in Tables 1 and 2 (crude extract and fractions respectively) reveal only weak or no bioactivity. Please, clarify and discuss your data.

We agree based on the break points and have deleted that statement. We have reviewed the publication you sent and have revised the discussion accordingly

Line 281-286

4. Line 342: This statement is true, but it needs a reference

A reference has been provided.

Line 347, reference [55]

5. Lines 360-361: The sentence “The reported activity justifies its use as an antifungal agent in the treatment of vulvovaginal candidiasis among the Ghanaian populace” must be deleted since it is not supported by data of the current study.

The statement has been deleted

CONCLUSION

1. Lines 368-371: This subject is not a conclusion of the current study, maybe is possible to include it in the discussion session

We have deleted that subject line

---

## [Decision Letter · Decision Letter 1]

22 Nov 2021

Bioactive triterpenoids from Solanum torvum fruits with Antifungal, Resistance Modulatory and Anti-Biofilm Formation Activities against Fluconazole-Resistant Candida albicans strains

PONE-D-21-19484R1

Dear Dr. Waikhom,

We’re pleased to inform you that your manuscript has been judged scientifically suitable for publication and will be formally accepted for publication once it meets all outstanding technical requirements.

Kind regards,

Horacio Bach

Academic Editor

PLOS ONE

Additional Editor Comments (optional):

Reviewers' comments:

Reviewer's Responses to Questions

**Comments to the Author**

1. If the authors have adequately addressed your comments raised in a previous round of review and you feel that this manuscript is now acceptable for publication, you may indicate that here to bypass the “Comments to the Author” section, enter your conflict of interest statement in the “Confidential to Editor” section, and submit your "Accept" recommendation.

Reviewer #1: All comments have been addressed

2. Is the manuscript technically sound, and do the data support the conclusions?

Reviewer #1: Yes

3. Has the statistical analysis been performed appropriately and rigorously? 

Reviewer #1: N/A

4. Have the authors made all data underlying the findings in their manuscript fully available?

Reviewer #1: Yes

5. Is the manuscript presented in an intelligible fashion and written in standard English?

Reviewer #1: Yes

6. Review Comments to the Author

Reviewer #1: I carefully read the authors' responses and changes made to the new version of the manuscript. I realized that the vast majority of suggestions were accepted, improving the quality of the text. Therefore, I consider it suitable for publication in PLOS ONE. Morevoer, in my opinion the manuscript meets all requirements and recommendations mentioned above.

7. PLOS authors have the option to publish the peer review history of their article (what does this mean?). If published, this will include your full peer review and any attached files.

Reviewer #1: **Yes: **Terezinha Inez Estivalet Svidzinski

---

## [Editor Report · Acceptance letter]

17 Dec 2021

PONE-D-21-19484R1 

Bioactive triterpenoids from *Solanum torvum* fruits with Antifungal, Resistance Modulatory and Anti-Biofilm Formation Activities against Fluconazole-Resistant *Candida albicans* strains 

Dear Dr. Waikhom:

I'm pleased to inform you that your manuscript has been deemed suitable for publication in PLOS ONE. Congratulations! Your manuscript is now with our production department. 

Kind regards, 

on behalf of

Dr. Horacio Bach 

Academic Editor

PLOS ONE